# Genome-Wide Identification of the *TGA* Gene Family and Expression Analysis under Drought Stress in *Brassica napus* L.

**DOI:** 10.3390/ijms25126355

**Published:** 2024-06-08

**Authors:** Yi Duan, Zishu Xu, Hui Liu, Yanhui Wang, Xudong Zou, Zhi Zhang, Ling Xu, Mingchao Xu

**Affiliations:** 1College of Life Sciences and Medicine, Zhejiang Sci-Tech University, Hangzhou 310018, China; d18846187062@163.com (Y.D.); xuzishu0613@163.com (Z.X.); 2Institute of Agriculture, The University of Western Australia, Crawley, WA 6009, Australia; hui.liu@uwa.edu.au; 3Leshan Academy of Agricultural Sciences, Leshan 614000, China; wwyyyhhh@163.com (Y.W.); z18080661800@163.com (X.Z.); z13990641077@163.com (Z.Z.)

**Keywords:** *Brassica napus* L., TGA transcription factor, bZIP, expression patterns, gene family

## Abstract

TGA transcription factors belong to Group D of the bZIP transcription factors family and play vital roles in the stress response of plants. *Brassica napus* is an oil crop with rich economic value. However, a systematic analysis of *TGA* gene family members in *B. napus* has not yet been reported. In this study, we identified 39 full-length *TGA* genes in *B. napus*, renamed *TGA1*~*TGA39*. Thirty-nine *BnTGA* genes were distributed on 18 chromosomes, mainly located in the nucleus, and differences were observed in their 3D structures. Phylogenetic analysis showed that 39 *BnTGA* genes could be divided into five groups. The *BnTGA* genes in the same group had similar structure and motif compositions, and all the *BnTGA* genes had the same conserved bZIP and DOG1 domains. Phylogenetic and synteny analysis showed that the *BnTGA* genes had a close genetic relationship with the *TGA* genes of the *Brassica juncea*, and *BnTGA11* and *BnTGA29* may play an important role in evolution. In addition, qRT-PCR revealed that three genes (*BnTGA14/17/23*) showed significant changes in eight experimental materials after drought treatment. Meanwhile, it can be inferred from the results of drought treatment on different varieties of rapeseed that the stress tolerance of parental rapeseed can be transmitted to the offspring through hybridization. In short, these findings have promoted the understanding of the *B. napus TGA* gene family and will contribute to future research aimed at *B. napus* resistant breeding.

## 1. Introduction

Plants are continuously subjected to an extensive array of biological and environmental pressures and have elaborated a variety of sophisticated defense mechanisms and complex signal networks to quickly perceive these challenges and regulate the expression of genes to successfully protect themselves from damage [1,2,3,4]. Transcription factors (TFs), as essential components of signal transduction, are prominently associated with the metabolic balance, growth, development, and multiple defense pathways of plants [5,6]. The TFs specifically activate or inhibit target gene expression by specifically interacting with the DNA *cis*-regulatory element sequence in the target gene promoter containing the binding site of the transcription factor, and guide the expression synchronously [7,8]. The bZIP transcription factor family is characterized by a vast array of members across various organisms and holds great significance in the TF families [9]. According to the amino acid sequence similarity and protein structure within the bZIP domain, the bZIP family of *Arabidopsis thalianais* is sorted into ten distinct groups, namely A, B, C, D, E, F, G, H, I, and S [10]. TGA (TGACG motif-binding factor) TF is classified within the D subfamily of the bZIP transcription factor family [11]. TGAs are composed of a typical bZIP domain, a highly variable N-terminal region, and a relatively conserved C-terminal region [12]. Within these domains, the bZIP domain can interact with specific DNA, serving as both a DNA-binding domain and a nuclear localization signal. The N-terminus exhibits significant variability in amino acid sequence and length. On the other hand, the C-terminus shows relative conservation, containing two glutamine-rich regions, Q1 and Q2. Additionally, the C-terminal region of TGAs also encompasses the DOG1 domain (Delay of Germination 1), which lacks DNA-binding ability but participates in the regulatory processes of TGAs [11,12,13,14]. *TGA1a* was the pioneering *TGA* gene to be cloned in plants and had performed as a pivotal landmark for characterizing the *TGA* gene family [15]. Subsequently, additional TGA TFs were discovered in wide-ranging plant species [10,16,17,18]. Based on sequence similarity, 10 TGA TFs in *A. thaliana* can be divided into five subgroups [14]. Within the *TGA* gene family, Group I consists of *TGA1* and *TGA4*, which exhibit the highest similarity to *TGA1a* in tobacco. Group II comprises *TGA2*, *TGA5*, and *TGA6*, showing close relationships and functional redundancy. *TGA3* and *TGA7* form Group III, while Group IV is composed of *TGA9* and *TGA10*. Group V consists solely of *TGA8,* which is also known as *PERIANTHIA (PAN)* [11,19]. Growing evidence suggests the vital involvement of TGA transcription factors in various biological processes, including pathogen defense and plant development [17,20,21,22,23,24], as well as in response to different stresses [25,26,27]. In addition, the role of TGA TFs is closely related to different plant hormone signaling pathways, including JA and SA [14,28].

Important crops such as rice, wheat, cotton, rapeseed, and sunflower have high economic and nutritional value [29,30,31,32,33]. *Brassica napus*, a globally cultivated oil crop, holds substantial economic significance due to its high commercial value [34]. *B. napus* originated through natural hybridization between *Brassica rapa* (ArAr, 2n = 20) and *Brassica oleracea* (CoCo, 2n = 18) approximately 7500 years ago [34]. It exhibits adaptability to various climatic conditions worldwide [35]. *B. napus* ranks as the third-largest contributor to vegetable oil production globally, accounting for approximately 13% of the total edible oil output [36]. The cultivation of *B. napus* is affected by many biotic and abiotic factors, such as drought, heat, and infection of *Sclerotinia sclerotiorum* [35,37,38]. Therefore, analyzing the resistance mechanism of *B. napus* varieties and excavating new resistance factors constitute important functions in *B. napus* resistance breeding. TGA TFS have been extensively investigated in multiple species; however, knowledge regarding TGA TFs in *B. napus* remains limited.

In this study, we accomplished an analysis of the *TGA* gene family in cruciferous plants, focused on *B. napus*. The phylogenetic relationship, structural characteristics, conserved motifs, chromosome localization, *cis*-element, and expression pattern of the *BnTGA* gene family under drought stress were described and analyzed. The comprehensive assessment of the *BnTGA* genes in this study offers potential candidate genes for clarifying the molecular regulatory mechanism of *B. napus* in the process of biotic and abiotic stress, which can be used to assist in the breeding of resistance varieties for *B. napus.*

## 2. Results

### 2.1. Identification and Location of Chromosomes of BnTGA Genes in Brassica napus

In this study, the protein sequences of 10 *A. thaliana* TGAs were used as query sequences in a BLAST search against the *B. napus* genome database. As a result, a total of thirty-nine putative *TGA* genes were identified in the *B. napus* genome. These genes were found to have entire bZIP and DOG1 domains based on the BLAST results. And these relevant *BnTGA* genes were renamed from *BnTGA1* to *BnTGA39*. Thirty-nine *BnTGA* genes were located on the 18 chromosomes of *B. napus* (Figure 1). Detailed information about these genes, such as their full-length protein sequences, molecular weight (MW), putative subcellular localization, and isoelectric point (pI), is summarized in Appendix A. The length of the protein sequences for these thirty-nine genes varies from 198 to 570. The theoretical isoelectric points (pI) and molecular weight (kDa) of putative TGA proteins ranged from 22.87 (*BnTGA22*) to 65.51 (*BnTGA17*) kDA and 5.72 (*BnTGA18*) to 9.36 (*BnTGA33*), respectively. Additionally, the prediction results of protein subcellular localization showed that *BnTGA* genes were concentrated within the cell nucleus.

### 2.2. Conserved Motif, Conserved Domain, and Gene Structure Analysis of TGA Gene Family in Brassica napus

To analyze the evolutionary relationship of *BnTGA* gene members using *B. napus* and *A. thaliana,* we constructed a phylogenetic tree with the maximum likelihood (ML) method in TBtools V2.096 software for the full TGA protein sequence. Based on the conserved structural domains of AtTGA proteins, thirty-nine BnTGAs were classified into five subgroups, which are as follows: *BnTGA1*, *BnTGA2*, *BnTGA3*, *BnTGA5*, *BnTGA7*, *BnTGA16*, *BnTGA23*, *BnTGA25*, and *BnTGA36* belonged to Group I; *BnTGA4*, *BnTGA14*, *BnTGA17*, *BnTGA21*, *BnTGA24*, *BnTGA26*, *BnTGA34*, and *BnTGA35* to Group II; *BnTGA8*, *BnTGA9*, *BnTGA10*, *BnTGA15*, *BnTGA20*, *BnTGA22*, *BnTGA27*, *BnTGA30*, *BnTGA33*, and *BnTGA37* to Group III; *BnTGA18*, and *BnTGA39* to Group IV; and the rest to Group V. Except for the Group IV, the proportion of the other four groups is similar (Figure 2a). The results of the phylogenetic analysis (Figure 2a) were confirmed through the MEME online website (https://meme-suite.org/meme/ (accessed on 22 March 2023)). We detected 10 conserved motifs, ranging in length from 11 to 50 amino acids, in the thirty-nine BnTGA proteins, taking the AtTGA proteins as templates (Figure 2b and Appendix A). The quantity of motifs present in the TGA proteins ranged from three to nine, and all proteins exhibited the presence of three conserved motifs: motif 1, motif 2, and motif 5. This observation indicates that motifs 1, 2, and 5 demonstrate a high degree of conservation within the BnTGA proteins. In the same subgroup, the characteristics of motifs were highly consistent, including motif number, composition, and relative position. For example, the number and distribution characteristics of motifs of each member in Group I and IV were the same, and in subgroup V, the similarity of the characteristics of motifs reached 92%. The motif composition of members of the same subgroup was relatively conserved, which were more inclined to express identical or similar functions. Meanwhile, the distribution characteristics of motifs were different in different subgroups, such as motif 10 only existed in Group I, and motif 9 was only identified in Group III. And motif 9 and motif 10 exhibit a highly conserved amino acid sequence (Appendix A), indicating that these two motifs are likely to have important implications for the structural and functional properties of the corresponding protein members within their respective groups. The variation in motif distributions across different evolutionary branches can result in alterations in the structure of *TGA* genes. These structural changes play a crucial role in determining the function divergence among different evolutionary branches. DOG1 and bZIP domains were found in thirty-nine BnTGA proteins, and the relative positions of these domains were consistently observed across different sequences (Figure 2c), suggesting that the *BnTGA* genes share a conserved domain arrangement in their gene structure.

Analyzing the gene structure is crucial for unraveling the connection between genome evolution and function divergence. The exon count of *BnTGA* genes varies from 3 to 12, highlighting the diversity in gene structure within this gene family. Most subgroup members contained a range of exons between 7 and 11. The *BnTGA15* group contained the largest number of exons, a total of 12, while *BnTGA33* contained only 3 exons (Figure 3).

In summary, subgroup members were highly conserved in motifs and domain, which further validates the accuracy of subgroup classification in the phylogenetic analysis.

### 2.3. Secondary and Tertiary Structural Analysis of TGA Proteins in Brassica napus

Proteins serve various functions in organisms, such as regulating gene expression and catalyzing biochemical reactions. The structural integrity of proteins plays a crucial role in enabling them to fulfill their functions. The functional significance of proteins heavily relies on their intricate 3D structure, so we conducted predictions regarding the secondary and tertiary structure of the BnTGA proteins. The predominant types of secondary structure in the BnTGA proteins were the α-helix (48.74–75.23%), followed by the random coil (17.22–41.24%), the extended strand (3.99–12.44%) and β-turn (1.53–4.56%) (Appendix A). These results indicated that the dominant components of the secondary structure in the BnTGA proteins were α-helix and random coil, while β-turn and extended strand were relatively less abundant. By using AlphaFold2, the 3D structure of the BnTGA protein was predicted. The structure of the BnTGA family protein primarily consisted of α-helix and random coil, as demonstrated in Appendix A, which was consistent with the secondary structure prediction of the BnTGA protein. Based on the evolutionary relationship between *B. napus* and *A. thaliana* TGA full proteins, we classified 39 TGA proteins into five groups according to the classification of TGA proteins in *A. thaliana*. In particular, the majority of TGA proteins in Group I, Group II, and Group V exhibited high structural similarity in their 3D structures, suggesting that these proteins likely have redundant or comparable functions in *B. napus*. Some TGA proteins in Group III and Group IV showed significant structural differences compared to other proteins within the same subgroup, indicating that they may have distinct functions. Furthermore, variations in 3D structures were observed among proteins from different subgroups (Appendix A).

When proteins play important roles in complex organisms, they often require interactions with other proteins or molecules, leading to the formation of protein networks and signaling pathways, enabling proteins to accomplish intricate and coordinated cellular processes in organisms. To investigate the protein interaction network of *TGA* genes of *B. napus*, we employed the String web tool for functional protein interaction prediction. *A. thaliana* was selected as a suitable organism for comparison with *B. napus*. The predicted results showed that the core members of the interaction network were members of the *TGA* gene family. There are interactions between different TGA proteins, and TGA proteins also interact with other transcription factors (Figure 4). They primarily interact with NPR proteins and GRXC proteins. The interactions among these proteins collectively regulate various metabolic reactions in organisms.

It is worth noting that *PAN* (*TGA8*), as a core member of the protein interaction network, interacted with multiple proteins, such as NPR1, NPR3, NPR4, NPR5, NPR6, GRXC7, and GRXC8, etc. *PAN*, as a single subfamily, has been confirmed to have a significant impact on plant flowering and root growth. Based on the predicted protein interaction network results, we aimed to investigate the functions of the *PAN* gene further. We used purchased *A. thaliana* mutant seeds to study the response of *PAN* mutant to drought stress. Col-0 and *PAN* mutant were germinated on 1/2MS medium and 1/2MS medium supplemented with PEG6000. After 3 days, the germination rates of both seeds were recorded (Appendix A). Regardless of the presence of PEG in the medium, the germination rate of the *PAN* mutant was consistently lower than that of the Col-0. As the concentration of PEG increased, both the Col-0 and *PAN* mutant germination rates decreased, with the *PAN* mutant showing a more pronounced decrease. After 7 days of growth in the medium, the phenotypic changes of both materials were documented, with the *PAN* mutant exhibiting shorter root lengths compared to the Col-0, consistent with previous research findings (Appendix A). Following drought treatment, the root length of *A. thaliana* seedlings displayed a decrease with increasing PEG concentration, and the change was more evident at 30% PEG concentration. Particularly, the *PAN* mutant showed a significantly shorter root length than the Col-0 after drought treatment. From this, we inferred that the *PAN* gene plays a crucial role in plant drought response. *BnTGA18* and *BnTGA39*, belonging to the same subfamily as *PAN*, may also have important roles in plants’ responses to drought stress.

### 2.4. Phylogenetic Analysis of TGA Gene Family in Brassica napus

To investigate the evolutionary relationship among *TGA* genes in various species, based on protein homology and cluster analysis, this study employed TBtools V2.096 software to analyze the TGA protein sequences of *B. napus*, *Arabidopsis thaliana*, *Arabidopsis lyrata*, *Arabidopsis halleri*, and *Camelina sativa*, and constructed a phylogenetic tree (Appendix A). In total, 105 *TGA* genes were identified across the five species, including 10 *AtTGA*s, 39 *BnTGA*s, 10 *AIyTGA*s, 13 *ArhTGA*s, and 33 *CsaTGA*s. Multi-sequence alignment of protein sequences was performed, and the phylogenetic tree categorized the *TGA*s into five distinct subgroups: Group I, Group II, Group III, Group IV, and Group V (Figure 5). In these five subgroups, except that Group III contained only two *BnTGA*s, *BnTGA18* and *BnTGA39*, in the remaining Group I, Group II, Group IV, and Group V, the distribution of *BnTGA*s was relatively average, with 9, 8, 10, and 10, respectively. The five species selected were all belonging to the cruciferous family. Overall, the BnTGAs protein had the highest sequence similarity to the TGAs protein in *Camelina sativa*. Therefore, TGAs in *B. napus* may have similar functions to TGAs in *Camelina sativa*.

### 2.5. Synteny and Duplication Analysis of TGA Gene Family in Brassica napus

Gene duplication could cause the amplification of the gene family, and two primary forms of duplication are tandem and segmental duplications. To comprehend the amplification mechanisms of *BnTGA* genes, we employed MCScanX to explore gene duplication events in *B. napus*. Consequently, a total of 73 pairs of gene duplication events were identified (Figure 6a). For a deeper analysis of the evolutionary connection between *TGA* genes in *B. napus* and various other cruciferous species, we also investigated the syntenic relationship of *TGA*s across six different plants. We identified orthologous gene pairs between *B. napus TGA* and several other plants, namely *Arabidopsis thaliana*, *Arabidopsis lyrata*, *Arabidopsis halleri*, *Camelina sativa*, *Brassica juncea*, and *Brassica oleracea* (Figure 6b). *B. napus* and other six species have gene pairs reaching 54 pairs with *Arabidopsis thaliana*, 53 pairs with *Arabidopsis lyrata*, 49 pairs with *Arabidopsis halleri*, 170 pairs with *Brassica juncea*, 106 pairs with *Brassica oleracea*, and 162 pairs with *Camelina sativa*. The findings indicated that the *TGA*s in *B. napus* exhibited a close genetic relationship with those in *Brassica juncea* and *Camelina sativa*, while showing distant phylogenetic relationships with *Arabidopsis halleri*, *Arabidopsis lyrata*, and *Arabidopsis thaliana*. Notably, *BnTGA11* and *BnTGA29*, reaching 26 gene pairs in *B. napus* and other six species, displayed significant homology with other species (Appendix A). This suggests that these two genes may serve as a major factor in the evolution of the *TGA* gene family.

### 2.6. Cis-Element in the Promoter Region of TGA Gene Family in Brassica napus

The *cis*-elements play a vital role in controlling transcription initiation and gene expression. To explore the transcriptional regulation of *BnTGA* genes, the *cis*-elements within their promoter regions were analyzed (Figure 7). The analysis of *cis*-elements revealed significant variations in the types and quantities of core components present in *BnTGA* genes (Appendix A). A total of 36 distinct *cis*-elements were identified to have potential functions. Among these *cis*-elements, 20 were related to growth and development, 6 were associated with stress reactions, and 10 were involved in hormone reactions. Most *BnTGA* genes contained Box 4 elements, G-box elements, TCT-motif, ARE, and ABRE elements. Among all *TGA* genes, the Box 4 element was the most abundant element, followed by the ARE core element. The SARE element was only included in *BnTGA33*. The *BnTGA2* promoter contained 18 *cis*-elements, which was the most diverse of all genes, but the *BnTGA24* promoter exhibited the lowest diversity in terms of *cis*-elements, with a mere 7 elements identified. Among the *cis*-elements of the *BnTGA* genes, the number of *cis*-elements that respond to different hormones varied greatly. The number of elements that respond to jasmonic acid and abscisic acid was similar, but the number of *cis*-elements that respond to salicylic acid was less than that of jasmonic acid and abscisic acid. The variations in hormone reaction elements suggested the significance of *BnTGA* genes in hormone signaling pathways. Furthermore, the diverse types and quantities of *cis*-elements across different genes implied the potential involvement of *BnTGA* genes in plant responses to biological and environmental pressures, as well as the regulation of environmental adaptation.

### 2.7. Expression Patterns of TGA Gene Family in Brassica napus

Examining the expression patterns of *BnTGA* genes offers valuable insights into their regulatory mechanisms. By studying RNA-Seq data that has been published in the NCBI database, we assessed the expression levels of *BnTGA* genes during various stress conditions. Specifically, under heat stress, all *BnTGA* genes exhibited increased expression, except for *BnTGA5*, *BnTGA7*, *BnTGA16*, and *BnTGA25*, which displayed lower expression compared to the control group, among which the transcription levels of *BnTGA4* (2.37-fold), *BnTGA14* (8.37-fold), *BnTGA28* (2.47-fold) and *BnTGA34* (4.00-fold) increased significantly (Figure 8a). Under drought stress, the expression of *BnTGA14* (2.11-fold), *BnTGA17* (1.54-fold), *BnTGA32* (1.84-fold), and *BnTGA35* (1.53-fold) increased significantly, while the expression level of the *BnTGA23* was markedly diminished (2.75-fold) in comparison to the control group (Figure 8a). Westar and ZY821 were *Sclerotinia sclerotiorum* susceptibility and resistance varieties, respectively. When the leaves of susceptible and resistant *B. napus* were infected by *Sclerotinia sclerotiorum,* respectively, most *TGA* gene expression levels had similar trends in susceptible and resistant *B. napus.* Notably, the expression of *BnTGA4*, *BnTGA6*, and *BnTGA11* in the Westar variety, infected by *Sclerotinia sclerotiorum*, was almost unchanged compared with the control group, while these three gene expression levels in the ZY821 variety were significantly reduced after *Sclerotinia sclerotiorum* infection (Figure 8b). The distinct expression levels of *BnTGA* genes under different conditions provide evidence for the involvement of *TGA* genes in the response of *B. napus* to both biotic and abiotic stresses, and it is speculated that *BnTGA4*, *BnTGA6*, and *BnTGA11* may be important genes for *B. napus* to respond to *Sclerotinia sclerotiorum*.

### 2.8. Expression Analysis of the BnTGAs under Drought Stress

Drought stress poses significant challenges to the growth, productivity, and quality of *B. napus*. To explore how *TGA* genes respond to drought stress in eight experimental materials, LY8, WJY520, LY07, LY1008H, DY6, ZS11, LY31AB, and ZD630, we subjected eight experimental materials to drought treatment. LY8 and WJY520 were crossbred using LY31AB as the female parent, while LY07 and LY1008H were used as the male parents in the crossbreeding (Figure 9). To facilitate phenotypic recording and photography, we assigned numerical labels to the experimental materials as follows: LY8, WJY520, LY07, LY1008H, DY6, ZS11, LY31AB, and ZD630 were labeled as 1–8, respectively. We selected three genes, *BnTGA14*, *BnTGA17*, and *BnTGA23*, which showed significant changes in expression levels in response to drought treatment according to transcriptomic data. We conducted qRT-PCR analysis to validate the results, and the findings were consistent with the transcriptomic analysis results (Figure 10a). Specific primers are shown in Appendix A. The expression levels of the three *BnTGA* genes after drought stress were significantly different. Following stress treatment, there was a noteworthy increase in the expression of *BnTGA14* and *BnTGA17* in the eight rapeseed species used in the experiment, whereas *BnTGA23* exhibited a markedly lower expression compared to the control group. This finding suggests that these three genes exhibit contrasting roles in response to drought stress. The increased expression of *BnTGA14* and *BnTGA17* may enhance the drought resistance of *B. napus*, while the increased expression level of *BnTGA23* may have an inhibitory effect. It is worth noting that although *BnTGA14*, *BnTGA17*, and *BnTGA23* showed significant changes compared to the control group after drought treatment, and their expression patterns were generally consistent among the eight experimental materials, LY8 exhibited the most superior performance among the eight experimental materials after drought treatment, presenting not only robust plant phenotypes (Appendix A) but also extremely significant expression differences in *BnTGA14* and *BnTGA17* (Figure 10b). In addition to LY8, LY31AB and ZD630 also demonstrated varying degrees of drought tolerance, but leaf wilting occurred in a small area along the leaf margins. Therefore, we speculate that LY8 may possess stronger adaptability under drought conditions, thereby promoting growth. The specific mechanisms underlying this phenomenon require further investigation.

## 3. Discussion

Transcriptional regulation is fundamental to various physiological processes in plants throughout their life cycle, including growth, development, and response to environmental stresses [39,40]. Gene regulation goes through a series of complex processes [41], in which transcription factors play an important role. TFs exhibit a wide range of functions in plant growth, including plant development and the formation of overall morphological diversity of plants [42,43]. Herein, gaining insights into the structure and function of TFs is essential to unravel the regulatory mechanisms that govern plant growth and development. TGA TFs play an important role in various physiological processes of plants, and many important studies have been carried out on their structure and function [14,44]. Nevertheless, there is a dearth of extensive studies on the *TGA* gene family of *B. napus*. In this study, we utilized the *AtTGA* genes as a reference to investigate the *BnTGA* gene family members throughout the entire genome. Additionally, we analyzed the gene structure and the gene evolutionary relationship of the *TGA* gene in different species. The expression patterns of the *BnTGA* genes were explored, including the differences in expression under heat and drought stress, as well as differences in expression after *Sclerotinia sclerotiorum* infection. This provides useful data for disease resistance breeding of *B. napus*.

Within this research, we identified 39 full-length *TGA* genes in *B. napus*. Through an analysis of gene structure and conserved motifs, we classified these TGA proteins into five distinct subgroups. Notably, this subgroup classification in *B. napus* aligns with the grouping observed in *A. thaliana* for the *TGA* gene family. Subgroup members were not only highly conserved in protein motifs, but also highly similar in gene structure [45]. Exon/intron and conserved motif analysis showed that there were significant differences in gene structure and sequence lengths of BnTGA members in different subgroups. It has been reported that introns can improve the content of mRNA by affecting transcription, and can also enhance mRNA translation efficiency [46]. The *BnTGA* gene may have different biological activities due to its different intron structures. Conserved domain and motif results showed the presence of typical motifs in all BnTGAs. High sequence similarity among motif sequences within each subgroup indicated that TGA members of each subgroup potentially share similar functions [47]. All BnTGA proteins contain bZIP and DOG1 domains, and relevant studies have demonstrated that the alkaline region within the bZIP domain utilized N-x7-R/K-x9, the fixed nuclear localization signal structure, to bind to DNA, thereby determining DNA specificity and nuclear localization [11]. To further explore their structural properties, we employed AlphaFold2 online website (https://alphafold.com/ (accessed on 16 May 2024)) for the prediction of the 3D structures of BnTGA proteins. The results indicated that the protein structure of 39 BnTGAs was in line with the secondary structure results and corroborated the phylogenetic classification results. The tertiary structure of members within the same subgroup is highly akin. Additionally, the protein interaction prediction results showed that the BnTGA*s* interacted with multiple proteins—primarily NPR proteins. This further confirmed the involvement of *TGA* genes in the NPR signaling pathway and it was consistent with the existing research results in other species [16,17,48]. Putative subcellular localization revealed that BnTGA TFs in *B. napus* were present in the cell nucleus, indicating that BnTGA TFs play a vital role in the cell nucleus. This indicated that although the functions of genes in each subgroup were different, the overall gene family functions remained conserved.

In evolutionary relationships, the sequences of proteins belonging to the same subgroup were highly similar, so they were very likely to perform similar functions. Based on sequence similarity, the *BnTGA* genes can be divided into five subgroups, which was consistent with *A. thaliana*. There were a total of 10 TGA TFs in the *A. thaliana* genome, and the classification and function of each member of the *AtTGA* have been thoroughly studied. The evolutionary relationship between *B. napus* and *A. thaliana* was utilized to infer the biological function of the *BnTGA* genes. It has been previously observed that *AtTGA01*-*AtTGA07* belong to three distinct subgroups; these seven *TGA* TFs play crucial roles in plant disease resistance [43]. Specifically, *AtTGA02*, *AtTGA05*, and *AtTGA06* serve as essential regulators in systemic acquired resistance (SAR) reactions, which are involved in plant defense against diseases [49,50,51]. Therefore, we hypothesize that *BnTGA6*, *BnTGA11*-*BnTGA13*, *BnTGA19*, *BnTGA28*-*BnTGA29*, *BnTGA31*-*BnTGA32*, and *BnTGA38*, through the NPR1 signal transduction pathway, may enhance plant disease resistance. Additionally, it has been observed that in the presence of cytokinin (CTK), *AtTGA03* interacts with ARR2 and binds to the PR1 promoter, thereby enhancing plant resistance against diseases [20,52]. Furthermore, *AtTGA01* and *AtTGA04* interact with the ethylene reaction factor (ERF) to enhance plant resistance [53]. It is speculated that *BnTGA1*-*BnTGA5*, *BnTGA7*, *BnTGA14*, *BnTGA16*-*BnTGA17*, *BnTGA21*, *BnTGA23*-*BnTGA26*, and *BnTGA34*-*BnTGA36* can also coordinate plant disease resistance through NPR 1 signaling pathways and other signaling pathways. *AtTGA09*, *AtTGA10*, and *AtPAN* play an important role in regulating the development of flower organs [54,55,56]. Moreover, studies have indicated a correlation between *AtPAN* and root growth [57]. It is speculated that the remaining *BnTGA* genes belonging to the same subgroup as *AtTGA09*, *AtTGA10*, and *AtPAN* may be related to flower organ development. Furthermore, members belonging to the same subgroup as *AtPAN* may be associated with root growth. Synteny analysis showed that among the six selected cruciferous species, BnTGAs had the closest genetic relationship and the most orthologous gene pairs with *Brassica juncea*, followed by *Camelina sativa*, but a more distant relationship with *Arabidopsis halleri*, so it can be postulated that the *TGA* genes in *B. napus* and *Brassica juncea* might demonstrate comparable functions, in line with the phylogenetic relationship between *B. napus* and other species. *BnTGA11* and *BnTGA29* display the highest number of homologous gene pairs across other species, suggesting their potential significance in *TGA* gene evolution [58].

The biological function of a gene product is often closely associated with *cis*-elements within its promoter region [59]. In this study, the analysis of *cis*-element showed the presence of diverse hormone-related *cis*-elements and stress-related *cis*-elements within the promoter region of the *BnTGA* gene family, including “WUN-motif”, “drought-inducibility”, and “defense and stress responsiveness”, etc. This showed that the *BnTGA* genes participate in the growth, development, and disease resistance of plants, through a variety of hormone-regulatory pathways and physiological processes that respond to stress. The different expression levels of the *BnTGA* genes under heat stress, drought stress, and *Sclerotinia sclerotiorum* infection further confirmed that the *BnTGA* genes play an important role in responding to stress. After the infection of *Sclerotinia sclerotiorum*, *BnTGA4*, *BnTGA6*, and *BnTGA11* changed significantly in resistant variety. Therefore, it is speculated that these *BnTGA* genes play an important role in responding to the infection of *Sclerotinia sclerotiorum*. In addition, the results of qRT-PCR were consistent with the trend of transcriptome data, and *TGA14*, *TGA17*, and *TGA23* changed significantly after drought treatment. It is further confirmed that TGA TFs play a vital role in the biotic stress response of *B. napus*.

The goal of rapeseed breeding is to develop varieties with good quality, high oil yield, and strong stress resistance. In preliminary experiments, we identified LY31AB and ZD630 as two excellent varieties. Using LY31AB as the maternal parent, we conducted hybrid breeding and obtained the first filial generation. To verify whether the progeny LY8 and WJY520 still possessed excellent traits and stress resistance, we subjected different rapeseed varieties to drought treatment and selected three genes with significant changes in the transcriptome data for validation. The qRT-PCR results showed that the LY8 exhibited strong stress resistance, manifested by its sturdy leaves in the phenotype, as well as significant changes in qRT-PCR data. Therefore, this variety is considered to be a superior variety, and further analysis of its stress resistance mechanism can be conducted in the future.

Overall, this study establishes the foundation for future study endeavors focused on unraveling the regulatory function of the *TGA* gene family in the growth and disease resistance mechanisms of *B. napus*. Furthermore, these results offer a valuable resource for utilizing the *BnTGA* gene family in resistance breeding programs for *B. napus*.

## 4. Materials and Methods

### 4.1. Plant Material and Stress Treatment

The *B. napus* seeds were from commercial varieties (DY6, ZS11 and ZD630) or our own groups (LY07, LY1008H, LY31AB, LY8 and WJY520). Seeds were germinated in a low-light environment, ensuring adequate moisture. Three days after seed germination, the seedlings were moved to a hydroponic tank filled with Hoagland nutrient solution, and the seedlings were kept growing at 24/20 °C (day/night), 14/10 h (light/night) and 60–70% relative humidity. To explore the response of TGA TFs to drought stress, twenty-day-old seedlings were immersed in Hoagland nutrient solution containing 10% PEG6000 to simulate drought treatment, with a treatment time of 12 h and 24 h, respectively. Additionally, 0.1 g of *B. napus* leaf samples were collected at 0 h, 12 h, and 24 h under drought treatment, respectively, and put into a 2.0 centrifuge tube marked with the sample number. The leaves were collected at the designated time point and subjected to three replicate samplings, immediately flash-frozen in liquid nitrogen, and then stored at −80 °C for subsequent analysis.

The *A. thaliana* Col-0 and *PAN* mutant seeds were purchased from AraShare (https://www.arashare.cn/index/Product/index.html), with Col-0 serving as the wild-type control. The seeds were surface-sterilized by treating them with 70% ethanol for 5 min, followed by 3% sodium hypochlorite for 5 min, and then rinsed with sterile water 10 times. The sterilized seeds were germinated on 1/2MS medium. Simultaneously, PEG6000 was added to the 1/2MS medium at concentrations of 10%, 20%, and 30% to simulate drought conditions [60]. *A. thaliana* seeds were vernalized for 24 h at 4 °C and then the 1/2MS medium was placed vertically in a greenhouse. After three days, the germination rates of *A. thaliana* Col-0 and *PAN* mutants under different treatments were recorded. After 7 days of growth, the phenotypic characteristics of seedling root growth under drought conditions were documented.

### 4.2. Identification of Members of the TGA Gene Family in Brassica napus

To identify *TGA* genes in *B. napus*, we utilized 10 *TGA* gene sequences (Appendix A) in *A. thaliana*, obtained from The Arabidopsis Information Resource (TAIR, http://arabidopsis.org (accessed on 10 March 2023)). The whole *B. napus* genome sequence was obtained from Ensemble Plants (https://plants.ensembl.org (accessed on 10 March 2023)) [61]. The AtTGA protein sequences were used as a query to implement a BLASTP search (E-value < 10 × 10^−5^) for the whole genome protein of *B. napus* to screen out the candidate *TGA* genes in *B. napus*. The BLAST search was completed in TBtools V2.096 [62]. Subsequently, the candidate proteins sequence were submitted to the SMART database (http://smart.embl.de/ (accessed on 20 March 2023)) and CDD database (https://www.ncbi.nlm.nih.gov/cdd/ (accessed on 20 March 2023)) websites, respectively, to check whether the putative *TGA* genes contained entire bZIP and DOG1 domains to further discern the *TGA* gene in *B. napus* [63,64]. The theoretical isoelectric point (pI) and molecular weight (MW) of the retrieved TGA protein sequence were predicted by the online tool ExPASy (https://web.expasy.org/protparam/ (accessed on 10 April 2023)) [32,65]. Subcellular localization of all retrieved TGA protein sequences was determined using (https://www.genscript.com/wolf-psort.html (accessed on 10 April 2023)).

### 4.3. Chromosomal Location, Gene Structure, Conserved Motif, and Conserved Domain Analysis of TGA Gene Family in Brassica napus

To validate whether the putative 39 *BnTGA* genes can be classified into the same five groups as the *TGA* genes in *A. thaliana*, the maximum likelihood (ML) method was used to analyze the gene clustering of *B. napus* and *A. thaliana* TGA proteins, with 1000 bootstrap replicates.

The positional arrangement and dispersion of the *TGA* genes on the *B. napus* chromosome were obtained based on the *B. napus* GFF file and the candidate *BnTGA* gene, utilizing the Gene Location Visualize from GTF\GFF function within the TBtools V2.096 software; all parameters were set to their default values according to the program [62].

Based on the downloaded *B. napus* genome annotation files and the candidate TGA protein sequence, the BnTGAs structure, motif, and conserved domains were depicted employing the Gene Structure View tool in TBtools V2.096; all parameters were set to their default values according to the program [62,66].

### 4.4. Protein Interaction, Secondary and Tertiary Structural Analysis of TGA Proteins in Brassica napus

Based on the *TGA* gene ID and protein sequences of *B. napus*, OrthoVenn3 (https://orthovenn3.bioinfotoolkits.net/ (accessed on 16 May 2024)) was used to compare the homologous relationship between *TGA* genes in *B. napus* and those in *A. thaliana*. The STRING website (https://cn.string-db.org/ (accessed on 16 May 2024)) was used to establish a protein-protein interaction network, selecting a confidence score of 0.4 for the interactome and keeping the remaining parameters unchanged at their default values. The maximum number of interactors to be displayed was limited to 20. Finally, Cytoscape 3.10.2 was utilized to visualize the protein-protein interaction network.

The secondary structure prediction of BnTGA proteins was completed using the SOPMA application (https://npsa.lyon.inserm.fr/cgi-bin/npsa_automat.pl?page=/NPSA/npsa_sopma.html (accessed on 16 May 2024)). To estimate and visualize the tertiary structures of the proteins, the Swiss-Model software (https://swissmodel.expasy.org/interactive (accessed on 17 May 2024)) was utilized [67,68].

### 4.5. Phylogenetic Analysis and Collinearity Analysis of TGA Gene Family in Brassica napus

To further understand the evolutionary relationships between BnTGA proteins and proteins from other plant species, including *Arabidopsis thaliana*, *Arabidopsis lyrata*, *Arabidopsis halleri*, and *Camelina sativa*, we constructed a phylogenetic tree. The *TGA* gene family in *Arabidopsis lyrata*, *Arabidopsis halleri*, and *Camelina sativa*, were identified according to the same method as *B. napus.* In addition, the TGA protein from *Arabidopsis thaliana*, *Arabidopsis lyrata*, *Arabidopsis halleri*, and *Camelina sativa* were obtained using the same methodology as in the case of *B. napus*. All TGA protein sequences in five species were multi-aligned through MUSCLE in MEGA (Version 11.0.13) [69]. Based on the JTT+G model, the phylogenetic tree was assembled using MEGA11 with default parameters, employing the maximum likelihood (ML) method with 1000 bootstrap replicates. The beautification of the evolutionary tree built using five species was completed by the Evolview online website (https://evolgenius.info//evolview-v2/#login (accessed on 26 April 2023)) [70]. The gene collinearity analysis of the *BnTGA* genes was carried out by MCScanX [71] to detect *TGA* gene replication events in *B. napus* with default parameters. In addition, six cruciferous plants (*Arabidopsis thaliana*, *Arabidopsis lyrata*, *Arabidopsis halleri*, *Camelina sativa*, *Brassica juncea*, and *Brassica oleracea*) were selected to construct a syntenic analysis map with *B. napus*.

### 4.6. Cis-Element Analysis in Promoters of TGA Gene Family in Brassica napus

The 2,000 bp upstream DNA sequences preceding the start codon (ATG) of the *BnTGA* genes were retrieved using TBtools V2.096. These sequences were then submitted to the PlantCARE database (http://bioinformatics.psb.ugent.be/webtools/plantcare/html/ (accessed on 4 June 2023)) to identify and retrieve *cis*-regulatory elements (CREs) within the promoter region of *BnTGA* genes [66]. TBtools V2.096 was employed for visualizing the distribution data of the *cis*-elements associated with *BnTGA* genes.

### 4.7. Expression Profile Analyses of TGA Gene Family in Brassica napus

To investigate the expression of the *BnTGA* genes under drought, heat stresses, and under infection of *Sclerotinia sclerotiorum*, RNA-Seq data were acquired from the NCBI database (GSE156029 and GSE81545). GSE81545 was used to study the expression of *BnTGA*s after *Sclerotinia sclerotiorum* infection [72]. GSE156029 was used to study the expression of *BnTGA*s under drought and heat stress [73]. The generation of Advanced Heatmap Plots was carried out using the tools available on OmicStudio tools at https://www.omicstudio.cn (accessed on 28 June 2023).

### 4.8. RNA Extraction and Quantitative qRT-PCR Validation

Total RNA from *B. napus* was extracted with a FastPure Plant Total RNA Isolation Kit (Vazyme, Nanjing, China). RNA was used to synthesize cDNA through the HiScript Ⅲ First Strand cDNA Synthesis kit (+gDNA wiper) (Vazyme, Nanjing, China). The process referred to the manufacturer’s instruction and the reverse transcription product was diluted as a template. The qRT-PCR was conducted using a ChamQ Universal SYBR^®^ qPCR Master Mix (Vazyme, Q311, Nanjing, China), with three replicates per sample. Primer Blast (https://www.ncbi.nlm.nih.gov/tools/primer-blast/ (accessed on 28 July 2023)) was used to design primers, and *Actin* gene was used as the internal reference gene. The results of the relative expression for target genes were calculated according to the 2^−ΔΔCt^ method [74]. The quantitative data were compiled using Excel 2016 software, while statistical analysis and visualization were conducted using GraphPad Prism8.

## 5. Conclusions

In this study, a total of 39 *BnTGA* genes were discovered and categorized into five subgroups after an extensive analysis of the *B. napus* genome. By analyzing the gene structure, conserved motif, conserved domain, phylogenetic relationship, *cis*-element, and 3D structures of the *BnTGA* gene, the biological characteristics of *BnTGA* genes were comprehensively evaluated. The expression differences of 39 *BnTGA*s were assessed in response to *Sclerotinia sclerotiorum* infection, leading to the identification of the *BnTGA4/6/11* genes as potential key players in the defense against *Sclerotinia sclerotiorum*. The qRT-PCR results demonstrated significant changes in the expression levels of three genes under drought stress: *BnTGA14* and *BnTGA17* exhibited a significant increase in expression, while *BnTGA23* displayed a considerable decrease. Meanwhile, by subjecting different varieties of *B. napus* to drought treatment, it can be inferred from the results that the superior traits and stress tolerance of parental rapeseed can be transmitted to the offspring through hybridization. These findings lay a solid foundation for future investigations exploring the regulatory roles of the *TGA* gene family in the growth and disease resistance mechanisms of *B. napus*. Furthermore, they offer valuable guidance for the potential application of the *BnTGA* gene family in resistance breeding programs for *B. napus*.

## Figures and Tables

**Figure 1 ijms-25-06355-f001:**
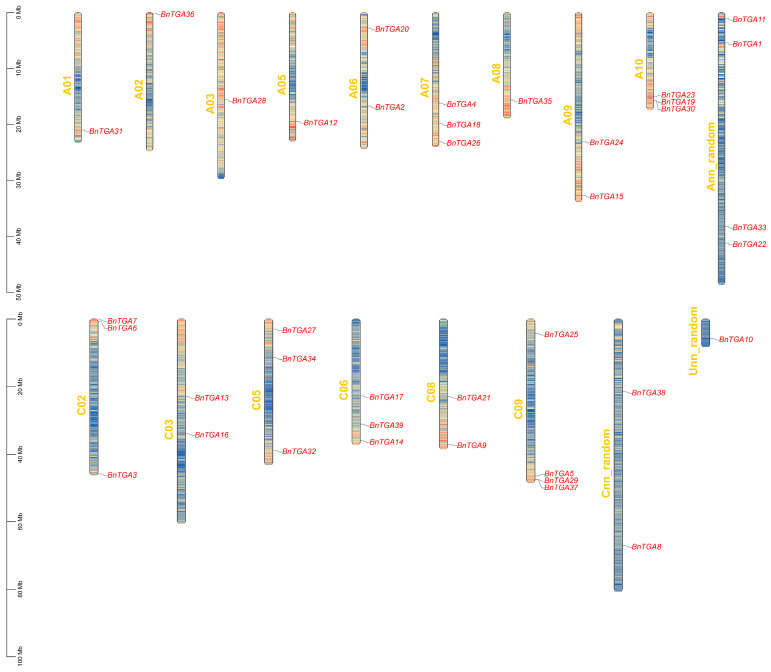
Distribution of *TGA* genes on *Brassica napus* chromosome. The chromosome size is expressed by length. The variation in color represents the density of the region, where blue indicates low, yellow indicates medium, and red indicates high density. The blue lines on the chromosome represent the physical location of the *BnTGA* genes. The scale on the left is displayed as megabases (Mb).

**Figure 2 ijms-25-06355-f002:**
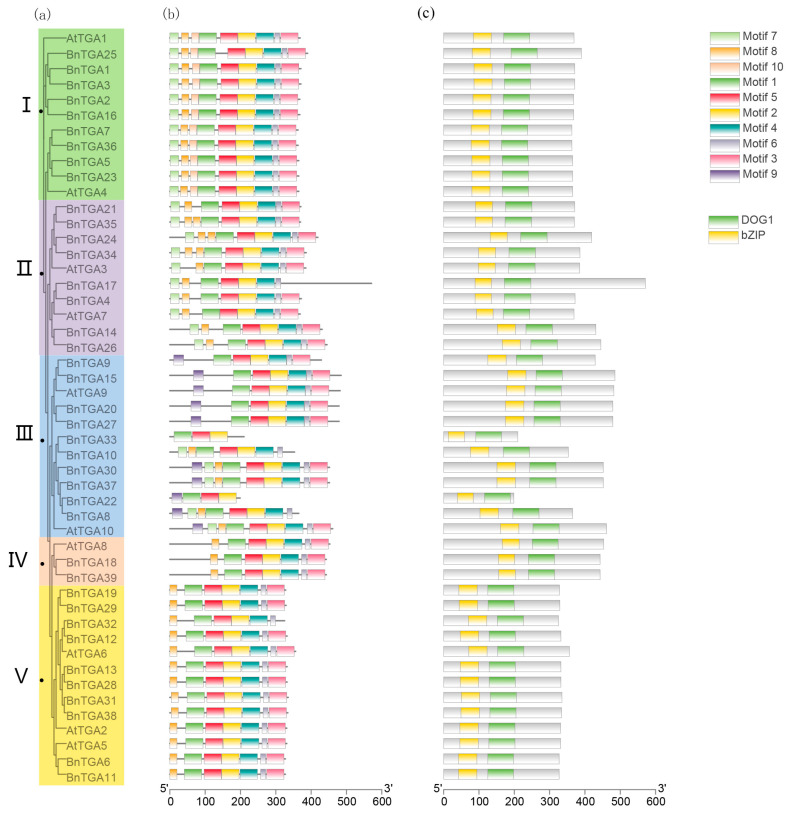
Phylogenetic relationship, motif, and domain of the *TGA* genes in *Brassica napus* and *Arabidopsis thaliana.* (**a**) The phylogenetic tree of the TGA protein sequence in *B. napus* and *A. thaliana* was constructed by the maximum likelihood (ML) method. The different groups were labeled from I to V. (**b**) Conserved motifs in BnTGAs. Various colors represented different motifs. (**c**) Comparison of conserved domain among AtTGAs and BnTGAs.

**Figure 3 ijms-25-06355-f003:**
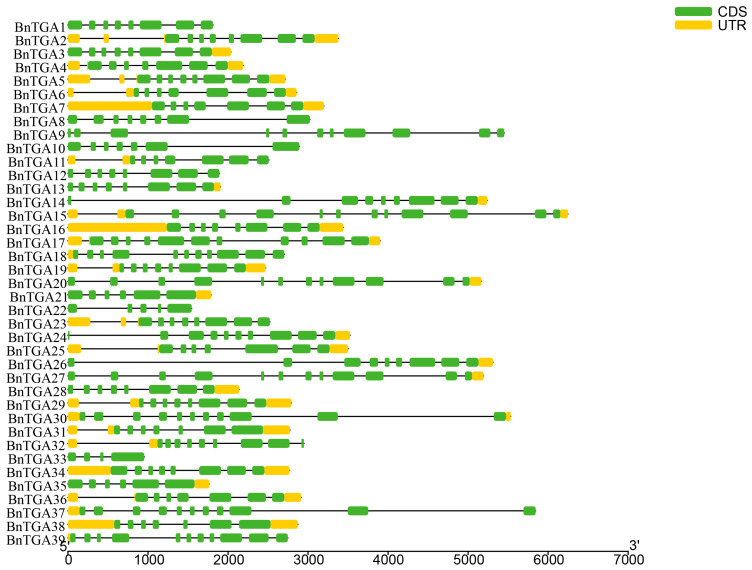
Thirty-nine *TGA* gene structures. The green region stands for the CDS, the yellow region for the UTR, and the gray lines refer to exon regions. The ruler at the bottom indicates the length of the sequences.

**Figure 4 ijms-25-06355-f004:**
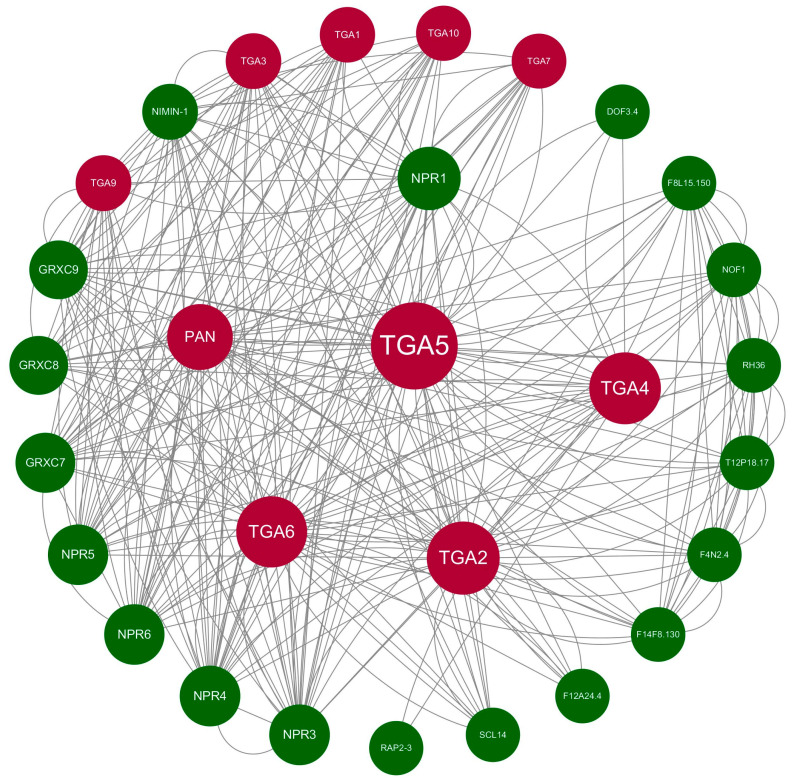
The prediction of the protein interaction network in *Arabidopsis thaliana*. Red represents TGA proteins, while green represents other proteins.

**Figure 5 ijms-25-06355-f005:**
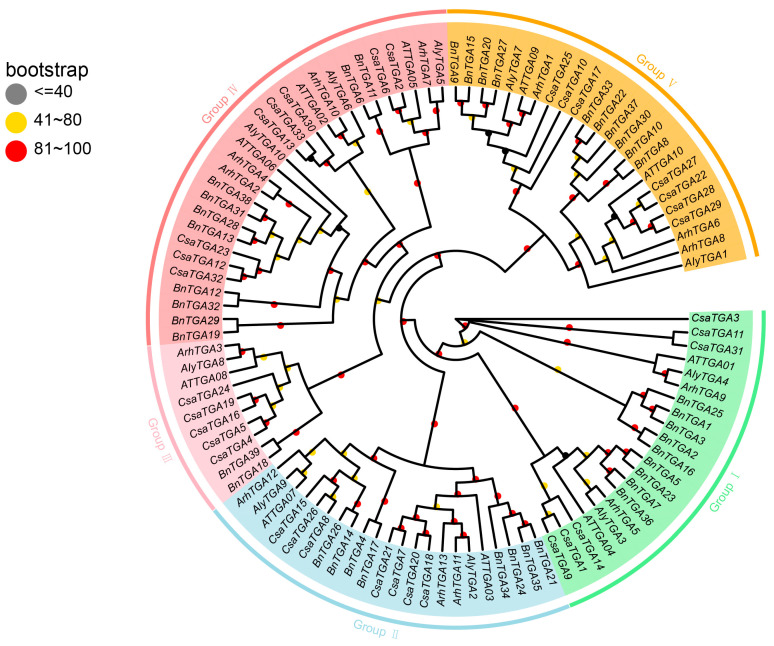
Evolutionary relationship analysis of TGA proteins. Based on the alignment of the TGA domain, TGA amino acid sequences from the *Brassica napus* (Bn), *Arabidopsis thaliana* (AT), *Arabidopsis lyrata* (AIy), *Arabidopsis halleri* (Arh), and *Camelina sativa* (Csa) were used to construct a phylogenetic tree, with the maximum likelihood (ML) method and 1000 bootstrap replicates. Light green, light blue, light pink, light coral, and orange represent Group I, Group II, Group III, Group IV, and Group V, respectively. Bootstrap values are symbolized with a circular pattern, with less than or equal to 40 in gray, 41 to 80 in yellow, and 81 to 100 in red.

**Figure 6 ijms-25-06355-f006:**
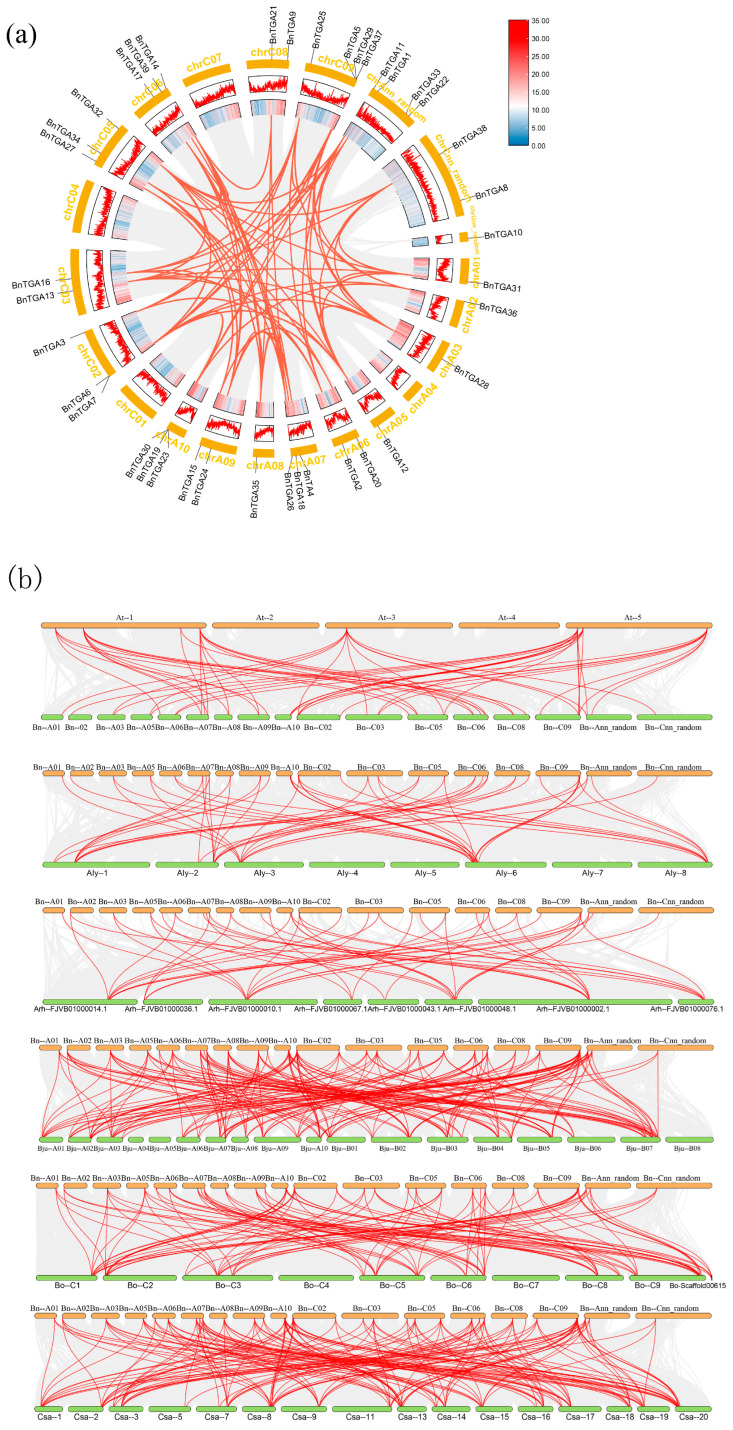
Gene duplication events of *BnTGA*s and synteny analyses of *TGA* genes among different plants. (**a**) Genome-wide gene syntenic analysis of *Brassica napus*. The gray lines represent the syntenic relationships of each gene in *B. napus*, while the red lines represent the replication events of the *BnTGA* genes. (**b**) Syntenic pairs of *TGA* genes between *B. napus* and other species (including *Arabidopsis thaliana*, *Arabidopsis lyrata*, *Arabidopsis halleri*, *Brassica juncea*, *Brassica oleracea*, and *Camelina sativa*). Gray lines in the background indicate the collinear blocks within *B. napus* and other plant genomes, while the syntenic *TGA* gene pairs are linked with red lines.

**Figure 7 ijms-25-06355-f007:**
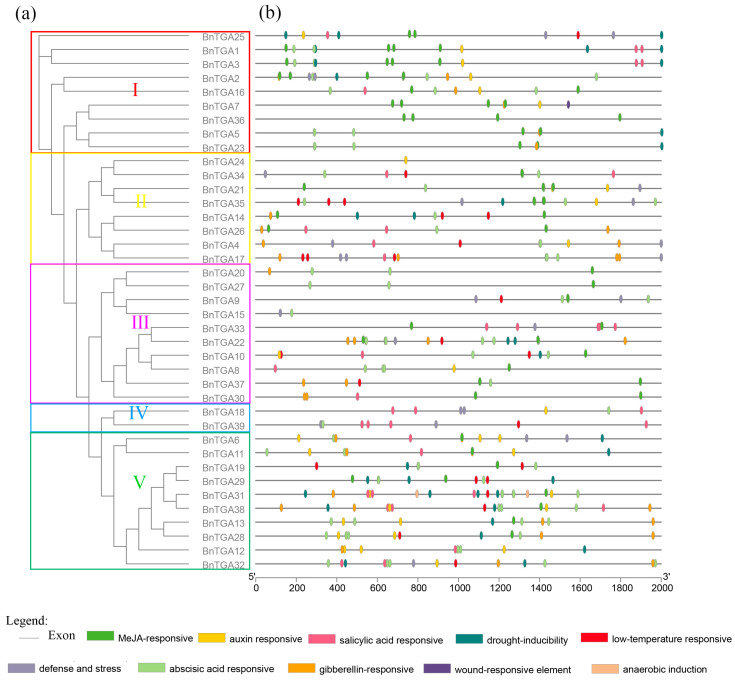
(**a**) Phylogenetic tree of the TGA protein sequence in *Brassica napus*. (**b**) *Cis*-acting element prediction of the *BnTGA* gene family in *B. napus*. Different colors represent the different *cis*-acting elements.

**Figure 8 ijms-25-06355-f008:**
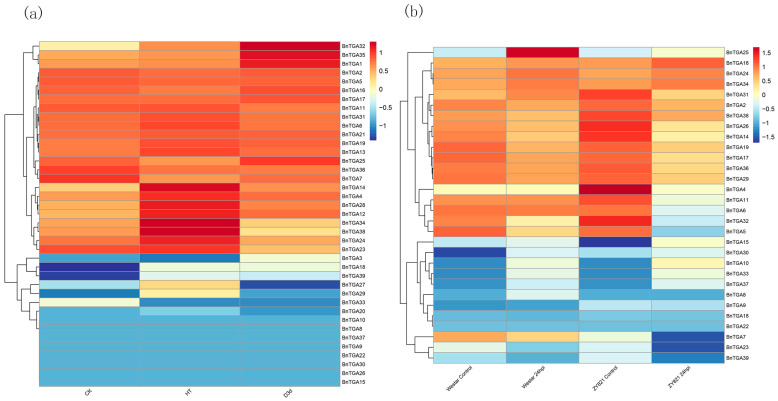
Expression analysis of *TGA* genes in *Brassica napus* leaves under biotic and abiotic stress. (**a**) Expression levels of *TGA* genes in *B. napus* leaves under drought and heat stress. (**b**) Expression levels of *TGA* genes in both susceptible (Westar) and tolerant (ZY821) genotypes of *B. napus* leaves infected with *Sclerotinia sclerotiorum*.

**Figure 9 ijms-25-06355-f009:**
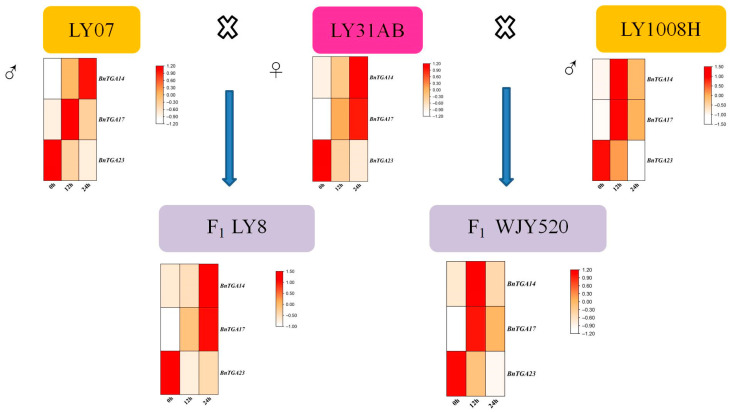
Schematic diagram of *Brassica napus* cross-breeding in this study.

**Figure 10 ijms-25-06355-f010:**
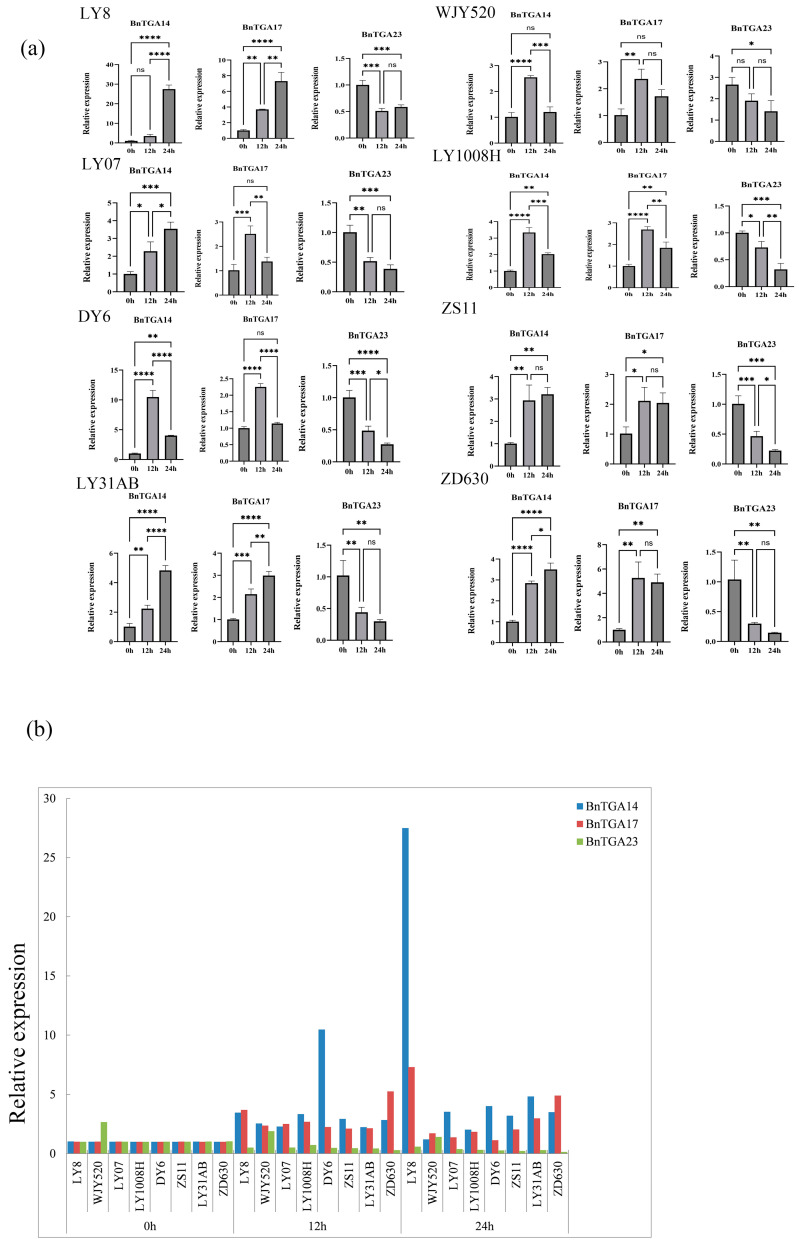
Analysis of *BnTGA* genes’ expression levels in *Brassica napus* under drought stress. (**a**) Relative expression levels of three selected *BnTGA* genes in eight experimental materials, at different time points, were measured by qRT-PCR in response to drought stress conditions. The expression levels were normalized to that of *Actin*. (*, *p* < 0.05; **, *p* < 0.01; ***, *p* < 0.001; ****, *p* < 0.0001; NS, no significant difference). (**b**) Summary of qRT-PCR expression of three *BnTGA* genes treated by drought in different varieties of *B. napus*.

## Data Availability

Data are available on request from the authors.

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
