# Peer review of "Genome-Wide Identification of the TGA Gene Family and Expression Analysis under Drought Stress in Brassica napus L."

_ijms, 2024, doi:10.3390/ijms25126355_

Round 1

Reviewer 1 Report

Comments and Suggestions for Authors

This study investigated the bioinformatics approaches of the TGA genes family and express analysis under drought stress in Brassica napus L. I think it is suitable for publication in the IJMS. My minor comments are as follows:

-check the title: express analysis or expression analysis or expressed?

In the overall manuscript (MS), please write genes and botanical names of species in italics and protein names non-italic

-in keywords, change Brassica napus L to Brassica napus L.

-keywords minimum five words

- Table S3 and S4 cited before S2, please arrange Table S1-S7 in main text

- figure quality is not good

- in the whole article, many places need space, for example, lines 137, 140, 190, 213, and so on

- In Figure 5 legend needs to explain B. napus (Bn)

- In Figure 5, legend needs space in B. napus, Arabidopsis thaliana(AT), Arabidopsis lyrata(AIy), Arabidopsis halleri(Arh), and Camelina sativa(Csa)

Reviewer 2 Report

Comments and Suggestions for Authors

In this article, the authors focu on the TGA gene family members in B. napus.

TGA transcription factor belongs to Group D of the bZIP transcription factors.

They identified 39 full-length TGA genes in B. napus

It could be an interesting paper

there are yet many concerns with the figures. They are often too small and. we cannot see

besides some methods are not well explained

line 16: what is DOG1 domain?

line 20: “materials”??? do you mean genes?

line 22/24: I do not get the point with AtPAN… is AtPAN a bZIP? what do you mean by “subfamily”?  are lines 22/25 points of Discussion? they should not appear in Abstract.

line 39; “encompassing a substantial size” what does that mean?

line 86: have you explained before what was DOG1 domain?

line 88: space is missing before (

line 90: do you mean putative subcellular localization?

figure 1 is hardly legible. Is yellow a good choice ? may be write your chromosomes on two lines

Figure 2 is also hard to read

I would split it in 2

line 111: space is missing before (

line 112: are the motifs detailed somewhere?

line 119: I think it is hard to comment on evolution if you study only one (2 with  Arabidopsis) species

line 167: you do not show any 3D structure?? are there differences in 3D structures between the clusters? what about motif 9? Ah yes it is in figure 3A. What is the point to make a figure with 3D structure (Fig 3a) and SPRING (Fig 3b)??

what can we see in 3a. blue on black is not the best choice? Is it relevant to put all the structures? Can not you make a choice to show and discuss the most relevant?

line 184: which proteins?  

Figure 3b. I do not understand. I am not used to SPRING. what did you use as query: Arabidopsis sequences or Brassica napus sequences? How can you get “Interaction

network of B. napus TFs and related functional proteins in Arabidopsis.”??? explain please

figure 3b is too small it is hard to see the colors of the lines

lines 180, 190, 196, 213, 217, 226, 316, 490 : space is missing before (

check everyhere

line 221: why did you use only Brassicaceae?

Figure 5? are there bootstraps? How many repetitions? are CsaTGA3, CsaTGA11, CsaTGA31 AtTGA01 AlyTGA4 and ArhTGA9 really group I? are they TGA?

line 232 : can you explain how MCScanx works?

line 272 and after: do you mean “jasmonic “?

figure 7: what is this tree? no explanation in caption….

Figure 10: maybe you can try to make it even smaller….

lines 507 to 514

go to new lines for each methods

what about bootstrapping?

why do we have “To classify BnTGAs, the maximum-likelihood (ML) method was used to analyze the gene clustering of B. napus and Arabidopsis TGA protein” lines 507/508 and 530 537?

 besides you did that on gene or protein? very confusing

Comments on the Quality of English Language

few mistakes to correct

Round 2

Reviewer 2 Report

Comments and Suggestions for Authors

My main concerns were with the figures. Unfortunately they did not take my remarks into account.

Figure 1. too small. Please do not put all the chromosomes on just one line. do it on two lines.. do you get what I mean?

caption: what is the colorcode for the chromosomes? what does that indicate?

Figure 2: split in two. Keep what is protein in one figure (tree, motif) and the gene structure in another figure

Figure 3. the 3D figures are not visible? What the point of showing them all. Do you discus their differences? show only one or one for each cluster

Split figure 3 into 2. There is no sense to have 3D structure and protein interaction in the same figure.

Figure 4: what about bootstrap values.? add evolution bar.

Figure 5: present in one column b under a; not b side by side to a

Figure 6. Enlarge. put the colorcode under and not side by side

Round 3

Reviewer 2 Report

Comments and Suggestions for Authors

The authors took into consideration my remarks that mostly concerned the figures

Yet figure 7 is still not legible. Do you think people can make the difference with the color code used for MEJA responsive and ABA responsive? bet-ween Auxin or gibberellin or anaerobic responsive? and so on…. change the colorcode

Round 4

Reviewer 2 Report

Comments and Suggestions for Authors

I am still not sure figure 7 is legible

some of the colors are very similar

maybe the best idea would be to have this figure as supplementary so people can zoom